# Computational assessment of the relationship between metabolism and histone methylation in cancer cells

**Mohammad Rasouli Koohi**, **Mahya Mehrmohamadi***

Department of Molecular Biotechnology, School of Biotechnology, College of Science, University of Tehran, Tehran, Iran

* mehrmohamadi@ut.ac.ir

## Abstract

Aberrant histone methylation and metabolic alterations are key hallmarks of cancer. Metabolic reprogramming during tumorigenesis could impact the histone methylation pattern by altering the availability of substrates and cofactors required for histone methyltransferases (HMTs) and demethylases (HDMs) activities. Despite advances in understanding this complex interplay, quantitative information about the contributions of specific metabolic shifts and histone methylation dynamics remains poorly understood. Here, we used multi-omics data integrated with machine learning models to discover key metabolites, genes, and pathways predictive of histone methylation levels in cancer cell lines. Our cell line models highlighted the significant role of metabolites associated with one-carbon, nucleotide, redox and lipid metabolism on histone marks. Validation in primary tumors confirmed the cell line models' findings. Overall, this study quantifies the contributions of the metabolic network to histone methylation variation in cancer cells.

## Introduction

Cancer cells show intricate epigenetics and metabolism differences with their normal counterparts and these alterations are classified as hallmarks of cancer [1]. Histone methylation plays a crucial role in cancer cells as a key epigenetic modification [2]. Histone methyltransferases (HMTs) are responsible for transferring methyl groups on lysine and arginine residue, while histone demethyltransferases remove methyl groups and reverse the process [3]. The metabolic network not only provides energy and building blocks for cells, but also generates metabolites that serve as substrates or cofactors for chromatin modifying enzymes. For instance, one-carbon metabolism and the methionine cycle play a crucial role in histone methylation through generating S-Adenosyl methionine (SAM), a universal methyl donor [4]. Methionine overconsumption in cancer cells enhances methylation of H3K4me3 by elevating SAM production for SET domain containing 1A/1B histone lysine methyltransferase

**Data availability statement:** The data underlying the results presented in the study are available from Cancer Cell Line Encyclopedia (CCLE) database (https://portals.broadinstitute.org/ccle). The computational code used for data analysis and model development has been made available on GitHub at https://github.com/mohammad-rasouli/Cancer-Metabolism/tree/main.

**Funding:** The author(s) received no specific funding for this work.

**Competing interests:** The authors have declared that no competing interests exist.

(SETD1A/B) [5]. Conversely, methionine restriction, either by dietary interventions or gene therapy reduces SAM levels, and increases S-Adenosyl homocysteine (SAH) which acts as a competitor for HMTs, thereby directly decreasing H3K4me3, H3K9me2, and H3K27me3, downregulating some growth-promoting genes, and arresting cancer cells in S/G2 phase [6]. Emerging evidence suggests that, alteration of other metabolic pathways including nucleotide metabolism can impact SAM availability and change histone methylation landscape as well [7]. Furthermore, elevated levels of glycolysis may deplete nicotinamide adenine dinucleotide ($NAD^+$) pools, which serves as a cofactor for $NAD^+$-dependent histone deacetylases such as Sirtuins [8,9].

Despite these insights, several gaps remain in our understanding of the metabolism-epigenetics crosstalk. Particularly, most studies focused on the role of a single metabolite or metabolic pathway on specific histone marks, while quantitative investigation of the broader metabolic network's interplay with a wide range of histone methylation marks remains less explored [10–13]. In this study, we leveraged multi-omics data from Cancer Cell Line Encyclopedia (CCLE) to identify the most significant metabolite, genes and metabolic pathways affecting histone methylation levels across 23 different cancer types. Beyond cell lines, we also utilized primary tumor samples to both validate our findings and reveal differences in the behaviors of cancer cells in vivo and in vitro. Our objective was to (i) quantify the predictive power of metabolic features on histone marks, (ii) discover novel links, and (iii) elucidate tissue-specific interactions that could guide therapeutic targeting.

## Results

### Machine learning models predict histone methylation levels from 'multi-omics' information

The CCLE dataset is one of the most comprehensive and diverse multi-omic cancer datasets publicly available. We studied the CCLE dataset with the goal to quantitatively investigate the impacts of a wide range of metabolites and metabolic genes on histone methylation marks in cancers [14,15]. This database is unique in providing cellular metabolome, transcriptome, and histone modification profiles across more than 870 cell lines from 23 different human cancer types. From CCLE's targeted metabolomics dataset generated by liquid chromatography followed by mass spectrometry (LC-MS), we used all the 225 metabolites, while from the transcriptomic data, we used 1927 transcripts with metabolism-associated functions based on Kyoto Encyclopedia of Genes and Genomes (KEGG) and Reactome annotations (S1 Table). For training predictive models, we used the Random Forests Regression (RFR) algorithm due to its robustness in handling complex and high-dimensional data. We trained a separate model for each of the histone methylation marks available in this dataset, by using either the metabolome, the transcriptome, or the combination of both features as input for predicting variations in each histone mark (Fig 1A; Methods). In each model, we divided the data into training and test sets to assess how the models work on the unseen data beyond the training data, and further

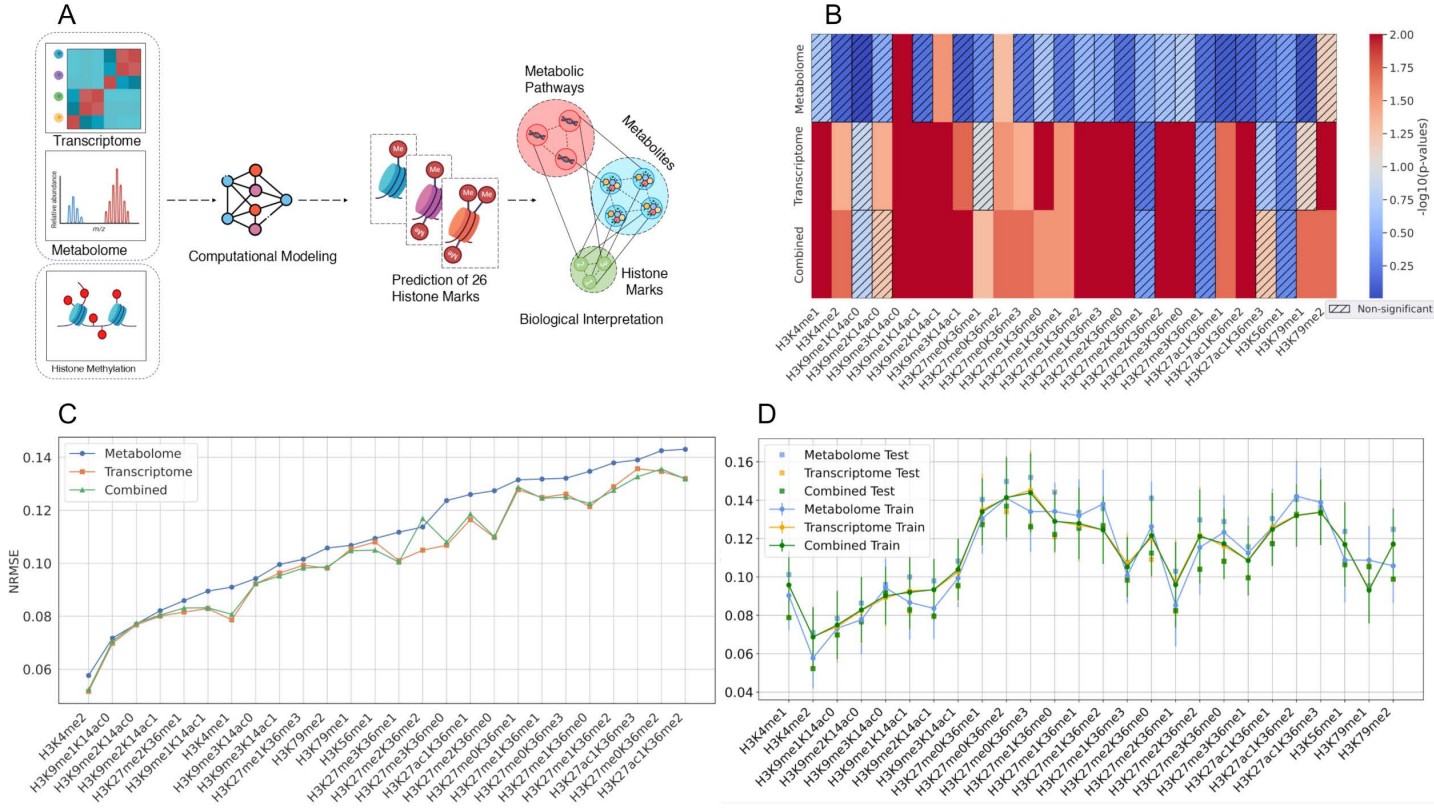

**Fig 1. Modeling and evaluation of histone methylation marks using multi-omics data.** (A) Histone methylation marks were predicted by training Random Forests (RF) models using metabolome, transcriptome, and combined datasets. Feature importance analysis identified key variables influencing histone methylation. (B) Predictability of 26 histone methylation marks using metabolome-only, transcriptome-only, and combined models based on permutation test p-values, with higher values indicating greater significance and hashed cells marking significant predictions. (C) Normalized Root Mean Square Error (NRMSE) on the y-axis was used to evaluate prediction accuracy for each histone methylation mark across the three models. (D) Model generalizability assessment by comparison of performance on training and test data. NRMSE values on cross-validation (CV) training data and test data across the three models used to assess the generalizability of each model to unseen data. A lower NRMSE on the test set indicates that the models effectively captured patterns in the data without overfitting.

divided the training data into 10 subsets (10-fold cross-validation). To identify histone marks that are predictable by the input feature sets, we employed a permutation test (see Methods). Finally, we calculated Normalized Root Mean Square Error (NRMSE) for predictions of each of these histone marks in each of the three models (i.e., metabolome, transcriptome, and the combined models) in order to assess the accuracy of the predictions and the generalizability of the models (see Methods).

Using RF regression, we trained models to predict histone methylation marks from metabolome-only (225 LC-MS metabolites), transcriptome-only (1,927 metabolism-associated genes from KEGG/Reactome), and combined (2,152 features) feature sets across 870 CCLE cell lines (Fig 1a; Methods). Our findings revealed that the metabolome-only models had weaker capacities to predict histone methylation on average, as indicated by their higher average NRMSE of 0.11 and their ability to predict only three histone methylation marks significantly accurately (Fig 1B, see Methods): H3K9me2K14ac0, H3K9me2K14ac1, and H3K27me0K36me2. In contrast, the transcriptome-only model outperformed the metabolome model by successfully predicting 19 histone methylation marks with a lower average NRMSE of 0.10. The combined model reflected the complementary contributions of metabolites and transcripts, as it was able to accurately

predict a broader range of histone methylation marks (20 in total) compared to either dataset alone. This indicates that integrating both omics layers captures complementary biological information that expands the model's predictive coverage (Fig 1B). Across various histone marks, H3K4me2 consistently showed the highest predictability across all three models, with NRMSE values of 0.051, 0.052, and 0.057 for the transcriptome, combined, and metabolome models, respectively (Fig 1C). To further validate our models, we compared the NRMSE from the 10-fold cross-validation training set with the NRMSE obtained from an independent test set. Nearly all histone marks exhibited lower NRMSE values for the test set, indicating that the models were effective in predicting unseen data without overfitting (Fig 1D).

### Identification of key metabolites influencing histone methylation

Based on the results from the metabolome-only models, we assessed the contribution of each individual cellular metabolite to the variation in each histone methylation. To achieve this, we initially identified histone methylation marks that were significantly accurately predictable based on the p-values derived from the permutation test as explained in the methods section. We next obtained feature importance scores for each of these histone marks, which are calculated by Random Forests reduction in model performance when the values of a particular feature are randomized (Fig 2a). We observed that cellular levels of multiple metabolites can significantly explain variations in H3K9me2K14ac0, H3K9me2K14ac1, and H3K27me0K36me2 histone methylation marks. The top metabolites contributing to the prediction of these histone

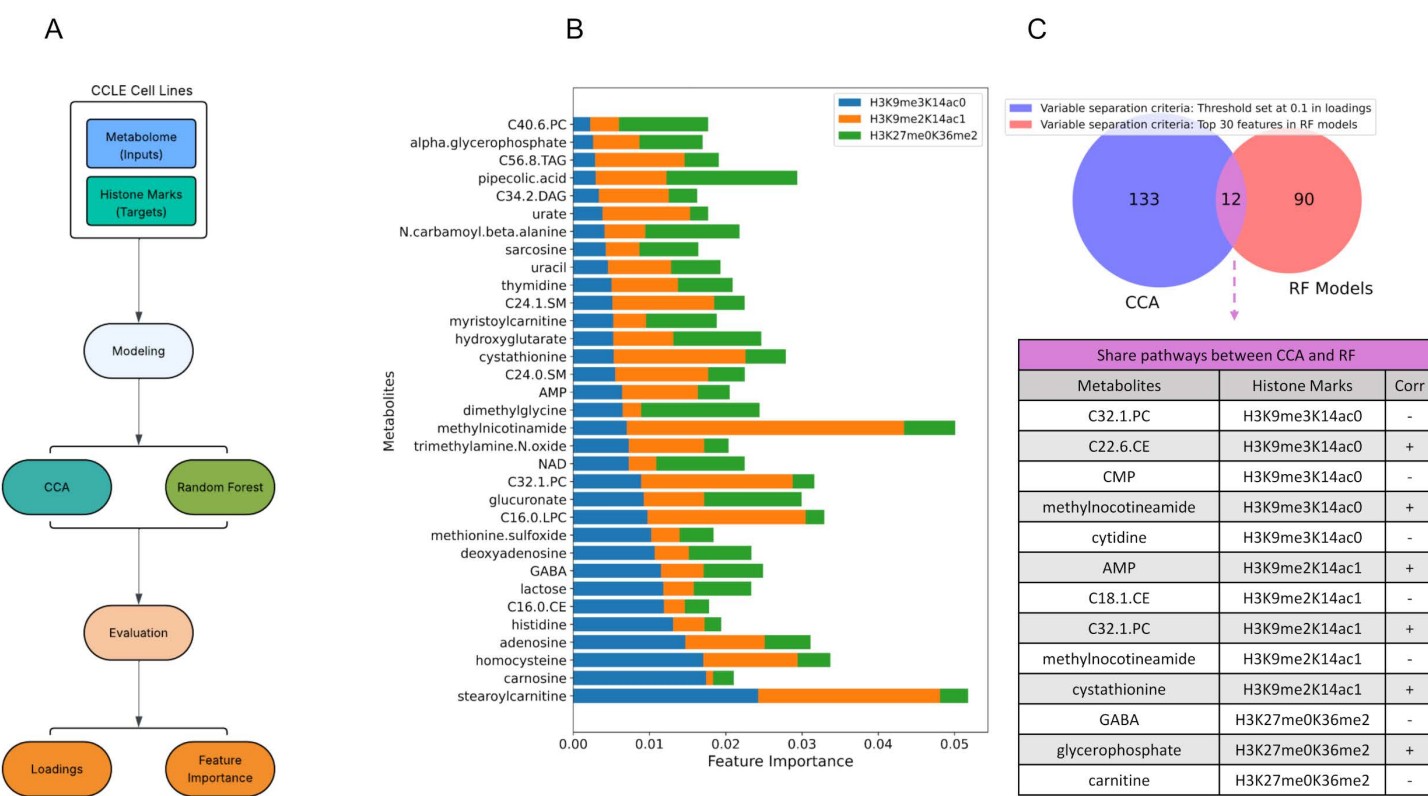

**Fig 2. Identification of metabolite associations with histone methylation marks.** (A) Workflow diagram of the pipeline for identifying significant metabolites associated with histone methylation. (B) Significant histone marks (based on permutation p-values) were selected from the metabolome-only model, and their top predictive metabolites are shown based on feature importance (x-axis). (C) The diagram shows overlapping associations identified by CCA (using a loading threshold of 0.1) and Random Forests (top 30 features based on importance) for significant histone marks, as listed in the table below the Venn diagram.

marks according to their importance scores are depicted in Fig 2B. These results indicate that metabolite groups such as one-carbon metabolism, nucleotide metabolism and energy carriers, and amino acid metabolism can influence histone marks through their direct association with one-carbon metabolism and the cellular methyl pool. However, beyond one-carbon metabolism, metabolites associated with lipid metabolism and redox metabolism also appear to play less well-characterized but potentially significant roles in explaining variation in histone marks. Particularly, one carbon related amino acids such as dimethylglycine (DMG), 1-methylnicotinamide (1-MNA), histidine, homocysteine, cystathionine, carnosine could alter the histone methylation landscape by directly affecting cellular methyl pools. Likewise, nucleotide metabolism, uracil, thymidine, adenosine monophosphate (AMP), adenosine, and NAD through their link with $NAD^+$/AMP pools have crucial roles in histone acetylation. For instance, nicotinamide n-methyltransferase (NNMT) as the key enzyme for 1-MNA production, can deplete the cells of SAM and inhibit HMTs [16]. Similarly, cystathionine as a key intermediate which links one-carbon metabolism to sulfur amino acid metabolism and redox balance, compete with methylenetetrahydrofolate reductase (MTHFR) by elevation of cystathionine β-synthase (CBS), and ultimately inhibit methionine and SAM regeneration [17]. SAM limitation could inhibit SET domain conformational change of G9 and SUV39H1, respectively responsible for H3K9me1/2 and H3K9me3 methylation [18,19]. Alpha-glycerophosphate and carnitine metabolism also directly affect sirtuin 1 (SIRT1), a key $NAD^+$-dependent deacetylase, by controlling $NAD^+$/NADH balance and indirectly regulate histone methyltransferases by modulating the chromatin state and availability of co-factors [20,21]. Beyond well characterized one-carbon metabolism and $NAD^+$ role in histone modifications, our models also identified lipid metabolism (e.g., stearoylcarnitine, myristoylcarnitine, phosphatidylcholines, sphingomyelins, diacylglycerol) and redox metabolism (e.g., alpha-glycerophosphate, glucuronate, urate, methionine sulfoxide) as a less known but significant metabolic pathways contributing to variations in histone modification levels.

Next, to complements the Random Forests feature importance analysis, which captures non-linear relationships between the metabolome and histone methylation, we employed canonical correlation analysis (CCA) to identify linear relationships between sets of metabolites that together affect sets of histone methylation marks (see Methods). This analysis revealed that only the first 6 out of the 26 canonical variates (CVs) have statistically significant p-values ($< 0.05$) (S2 Table). To identify the most influential variables, we used absolute loadings, which represent the weights assigned to each variable (metabolite or histone methylation mark) within a CV. Higher absolute loading values, considering both positive and negative signs, indicate a stronger contribution of the variable to the overall CV. After separating the top 30 features of the significant histone methylation marks from RF models, we applied a 0.1 threshold on loadings of significant CVs from CCA (Fig 2C). This process yielded 133 metabolites-histone methylation variable pairs (S1 Fig). By considering the same histone methylation mark-metabolite pairs, we identified 12 consistent relationships, including alpha-glycerophosphate, GABA, and carnitine for H3K27me0K36me2; 1-MNA, C32:1 PC, and cystathionine for H3K9me2K14ac1; and C32:1 PC, CMP, and cytidine for H3K9me3K14ac0 (S2 Fig). By integrating results from CCA and RF models, we meant to enhance the robustness of our findings and have a comprehensive view about the linear and non-linear relations between histone methylation and metabolite levels in cells. Together, our findings quantitatively show that metabolism plays a role in histone methylation in human cancers, and suggest that the link between certain metabolites, such as those found in carbon and lipid metabolism, with variation in histone methylation.

## Identification of key genes and pathways influencing histone methylation

We next aimed to look for transcripts whose expression levels were predictive of histone methylation levels in cancer cells. Similar to the steps described in the modeling above, we trained predictive RF models this time using transcript levels as input features, and then evaluated the models using permutation tests. We then obtained the most important features from the significant models. For each histone methylation mark, we performed a pathway enrichment analysis (PEA) on top selected transcripts in order to gain insight into the functional relationship of these features with histone methylation (Fig 3A). The results of extracting top important transcripts based on their feature importance values showed

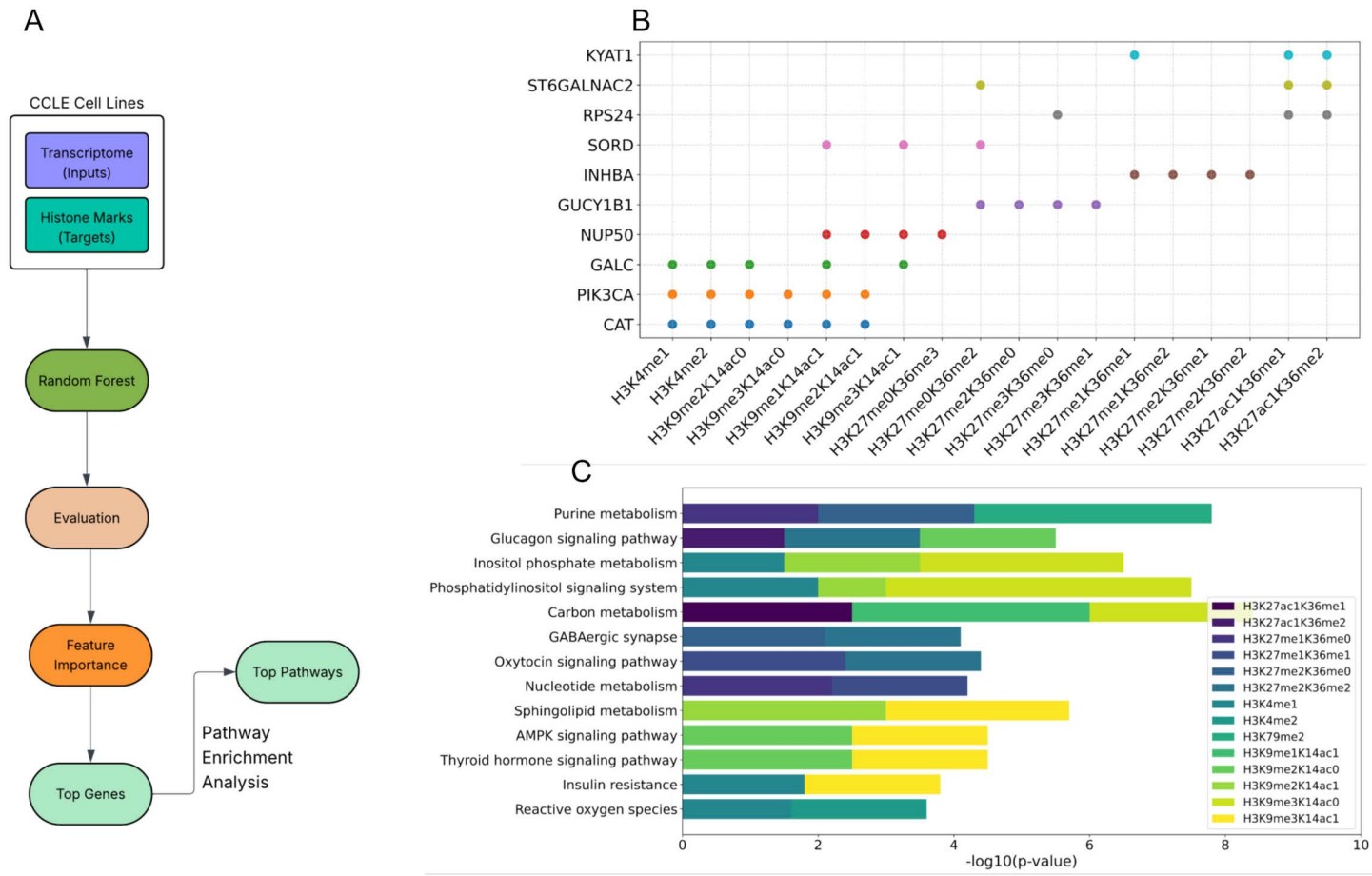

**Fig 3. Transcriptome-based identification of gene and pathway associations with histone methylation.** (A) Workflow diagram of the transcriptome model pipeline for identifying significant genes and metabolic pathways associated with histone methylation. (B) The scatter plot displays the top genes contributing to the prediction of significant histone methylation marks, as determined by their feature importance scores in the transcriptome model. (C) Pathway enrichment analysis of top genes associated with significant histone methylation marks. Top 30 genes based on the feature importance of the transcriptome model for each significant histone methylation mark were used for pathway enrichment analysis on KEGG to find the pathways associated with the methylation marks.

that the histone marks H3K4me1, H3K4me2, H3K9me2K14ac0, and H3K9me1K14ac1 share common top features. These features include catalase (*CAT*), phosphatidylinositol-4,5-bisphosphate 3-kinase catalytic subunit alpha (*PIK3CA*), and galactosylceramidase (*GALC*). H3K9me3K14ac0 and H3K9me2K14ac1 also have CAT and PIK3CA among the top features (Fig 3B). In order to have a better vision of the functionality of the top genes, we performed pathway enrichment analysis (see Methods). We observed that the AMP-activated protein kinase (AMPK) signaling pathway and the thyroid hormone signaling pathway were both enriched for H3K9me2K14ac0 and H3K9me3K14ac1 (Fig 3C). In addition, H3K4me1 and H3K4me2 histone marks were associated with the Reactive Oxygen Species (ROS) pathway expression. These findings are consistent with the previous research showing thyroid hormone (T3) as the key endocrine regulator of metabolism during lipophagy by raising oxidative phosphorylation and producing reactive oxygen species (ROS) [22]. Elevated ROS by modulating HMTs and HDM can affect histone methylation marks. For instance, it has been shown that the ROS, by influencing the SET1/MLL family of histone methyltransferases, can modulate H3K4 methylation. On the other hand, removing methyl groups from H3K4me1 and H3K4me2 by the Lysine-specific demethylase 1A (LSD1) enzyme

can further produce H2O2 as a byproduct [23]. In concordance with previous studies, that showed the remarkable role of carbon metabolism in the production of key metabolites involved in histone methylation [24], we identified this pathway as a major contributor to H3K9me1K14ac1, H3K9me3K14ac0, and H3K27ac1K36me1 histone methylation marks (Fig 3C). Moreover, the phosphatidylinositol signaling system and inositol phosphate metabolism pathways were the major predictors of H3K4me1, H3K9me2K14ac1, and H3K9me3K14ac0. PIK3CA, one of the top features identified, encodes the p110α catalytic subunit of phosphoinositide 3-kinase (PI3K), an enzyme that plays a crucial role in the phosphatidylinositol signaling system [25]. Together, these results illustrate that histone methylation is strongly influenced by gene expression of CAT, PIK3CA, phosphatidylinositol signaling system, and carbon metabolism.

## Combined effects of metabolite and transcript levels on histone methylation marks

To gain a comprehensive understanding of the combined effects of the metabolome and transcriptome on histone methylation, we integrated both datasets as input features. We then fed these features into new RF regression models for each histone methylation mark separately. After evaluating the models using feature importance values, we extracted the most significant features (transcripts or metabolites) associated with each histone methylation mark (Fig 4A). The results showed that, despite the significant contribution of metabolites, expression levels of genes exhibited much higher

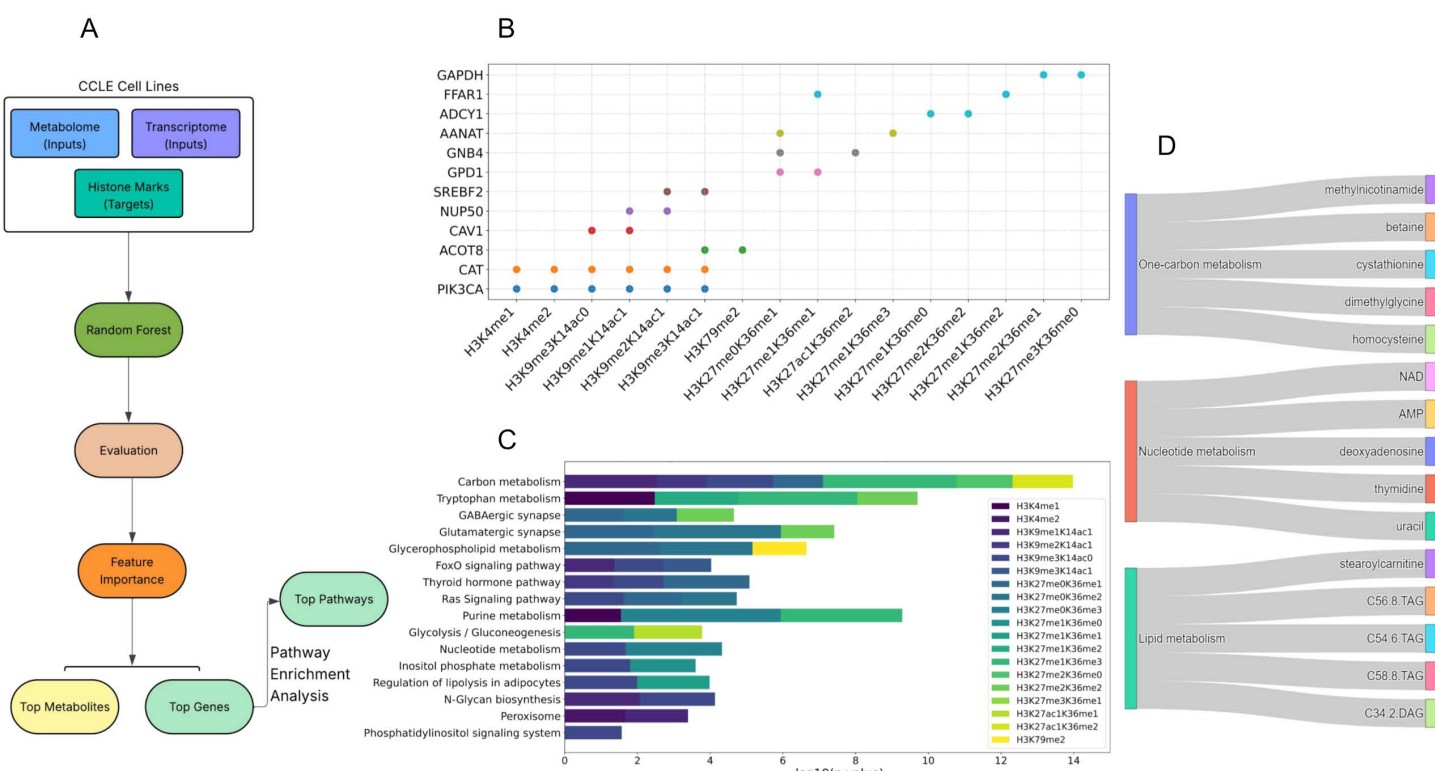

**Fig 4. Integrated analysis of genes, pathways, and metabolites driving histone methylation patterns.** (A) Workflow diagram of the combined model pipeline for identifying significant genes and metabolic pathways associated with histone methylation. (B) The scatter plot displays the top genes contributing to the prediction of significant histone methylation marks, as determined by their feature importance scores in the combined model. (C) Pathway enrichment analysis of top genes associated with significant histone methylation marks. Top 30 genes based on the feature importance of the combined model for each significant histone methylation mark were used for pathway enrichment analysis on KEGG to find the pathways associated with the methylation marks. (D) Top metabolites based on the feature importance from the combined model involved in one-carbon, nucleotide, and lipid metabolism significantly contributed to histone methylation prediction based on the feature importance of the combined models.

importance when integrated in the combined model. Similar to the transcriptome-only model, CAT (involved in ROS detoxification) and PIK3CA (encoding PI3K p110α) were dominant features in the combined model, which could be due to their roles as upstream regulators of metabolic flux and epigenetic enzymes (Fig 4B). After pathway enrichment analysis, we observed pathways including carbon metabolism, thyroid hormone signaling, purine metabolism, inositol phosphate metabolism, nucleotide metabolism, and the phosphatidylinositol signaling system, appeared to be the most significant pathways, confirming the findings of the transcriptome-only model (Fig 4C). In the combined model, carbon metabolism was able to explain a wider range of histone methylation marks compared to the transcriptome-only model (Fig 4c). Furthermore, the combined model revealed new pathways related to histone methylations, which were not identified in the transcriptome model. For example, tryptophan metabolism showed remarkable connection to various histone marks such as H3K4me1, H3K27me1K36me2, H3K27me1K36me3, and H3K27me2K36me2 (Fig 4C). Tryptophan metabolism generates $NAD^+$ through the kynurenine pathway, and $NAD^+$ is a key cofactor for $NAD^+$-dependent histone deacetylases such as Sirtuins. Changes in $NAD^+$ availability modulate Sirtuin activity, thereby influencing the balance between histone acetylation and deacetylation and reshaping the overall histone acetylation landscape [26].

From the metabolite features, $NAD^+$, 1-MNA, homocysteine, betaine, stearoylcarnitine, and triacylglycerols (TAG) were among the top predictors in the combined model (Fig 4D). Generally, metabolites identified with a strong connection with histone methylation in the combined model can be categorized into three main processes: one-carbon metabolism, nucleotide metabolism, and lipid metabolism (Fig 4D). The combined model confirms our previous findings. A key observation from our combined model was that despite the integration of both omics' layers, the feature importances and predictive powers were predominantly driven by transcriptomic features rather than metabolites. This aligns with our initial finding that transcriptome-only models outperformed metabolome-only models. While the combination of metabolome and transcriptome layers did not significantly enhance the prediction accuracy (NRMSE) for most marks, it enabled us to predict a wider range of histone marks, and identify novel associations such as tryptophan metabolism. This suggests that while gene expression provides the dominant signal for predicting histone methylation levels, metabolic features capture complementary biological information. Metabolites however, reflect the functional output of metabolic pathways and the availability of key co-factors (e.g., SAM, $NAD^+$). These can modulate histone methylation in a context-dependent manner that is not fully captured by transcript levels alone.

**Cell-type specific models characterize differences across cancers**

By considering all cancer types from the CCLE together in our models, we provided a holistic prospective on significant histone methylation marks and their association with metabolism. However, as epigenetic mechanism are highly tissue-specific, this approach may have limited our ability to assess cancer-specific relationships between metabolism and histone methylation. To address this, we next constructed cancer-type-specific models for cancer types with sufficient sample sizes in the CCLE dataset, namely lung (164 samples), hematopoietic and lymphoid tissue (158 samples), skin (47 samples), and breast (48 samples) cancers. In both lung and hematopoietic/lymphoid cells, H3K27me1K36me0, H3K27me1K36me2, H3K79me1, and H3K79me2 showed strong associations with metabolism, suggesting a significant tissue-specific role of H3K27 and H3K79 methylation marks in these cancer types (Fig 5A). No significant models were obtained for skin and breast cancers, possibly due to limited samples sizes. Feature importance analyses on lung and hematopoietic/lymphoid models revealed that these shared histone marks were associated with overlapping metabolic pathways across both cancer types, including purine metabolism (H3K27me1K36me2), glycosaminoglycan biosynthesis (H3K27me1K36me0), and biosynthesis of cofactors (H3K79me2). However, H3K9me2K14ac1, H3K27me0K36me2/3, H3K27me2K36me1, and H3K27ac1K36me1/2 were uniquely found significant in lung cancer, while H3K4me1/2, H3K27me1K36me1, and H3K56me1 were enriched in hematopoietic/lymphoid tissue cancer.

In lung cancer, top transcripts were enriched in neurotransmitter-related pathways such as GABAergic, glutamatergic, and cholinergic synapses (H3K9me2K14ac1), as well as cysteine and methionine metabolism (H3K27me0K36me2),

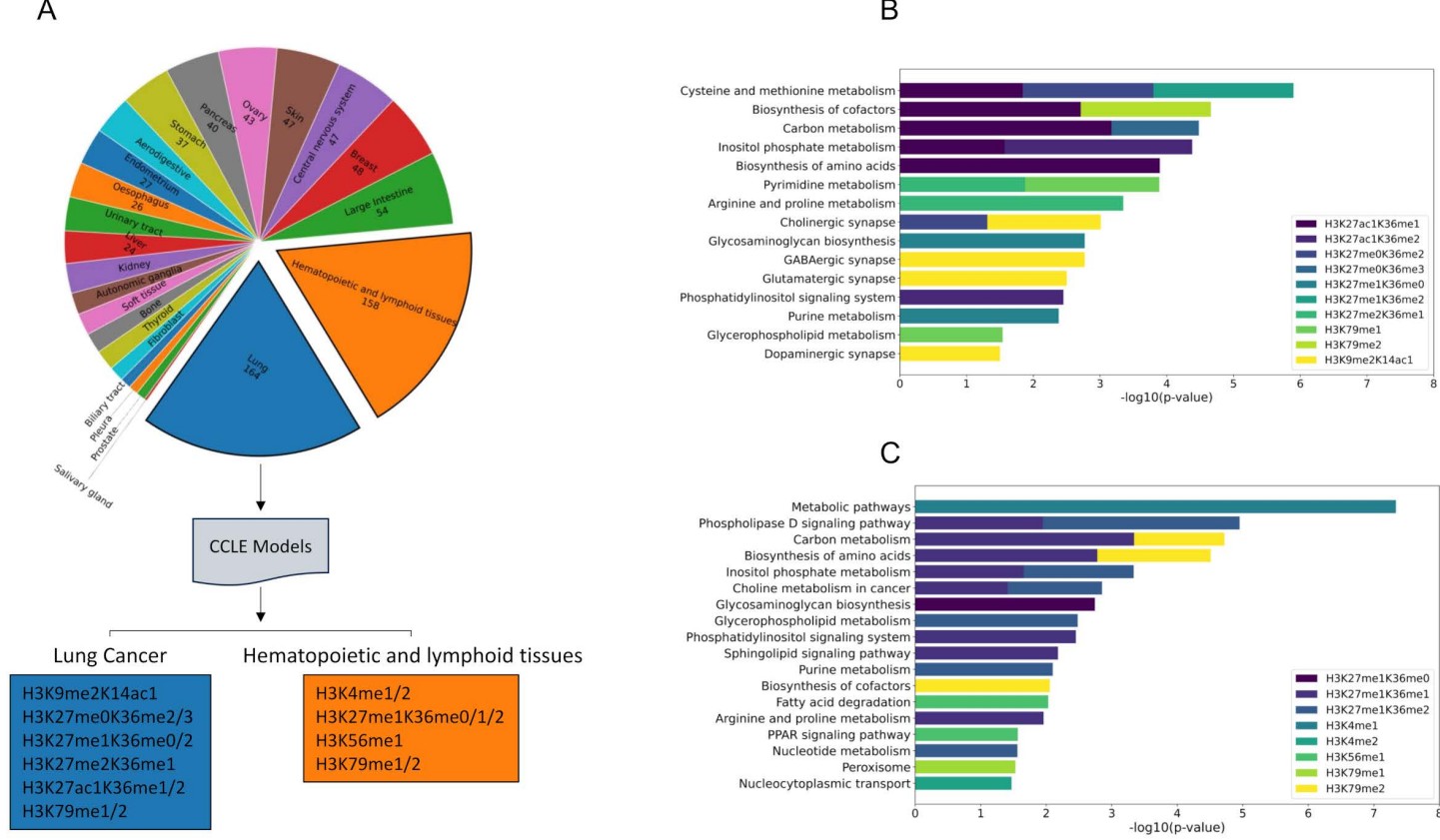

**Fig 5. Cancer-type specific metabolic predictors of histone methylation patterns.** (A) Cancer-type specific models built for skin, hematopoietic/lymphoid, breast, and lung cancer cells identify histone marks predictable by metabolism in each cancer types. (B) Top metabolic pathways including cysteine and methionine metabolism, biosynthesis of cofactors, and carbon metabolism predict histone methylations in lung cancer. (C) Phospholipase D signaling, carbon metabolism, biosynthesis of amino acids, and inositol phosphate metabolism demonstrated strong association with significant histone methylation marks in hematopoietic and lymphoid tissue cancer cells.

carbon metabolism (H3K27me0K36me3), and inositol phosphate signaling (H3K27ac1K36me2) ([Fig 5B]). In contrast, unique marks in hematopoietic/lymphoid tissue cancers were associated with broader metabolic processes, including fatty acid degradation and PPAR signaling (H3K56me1), as well as core pathways like biosynthesis of amino acids and phospholipase D signaling (H3K27me1K36me1) ([Fig 5C]).

## Validation in primary tumor samples

While our CCLE-based models revealed a strong association between metabolism and histone methylation, these findings may not fully capture the molecular interactions occurring in primary tumors, as cell lines undergo many adaptive changes during in vitro culture. To address this, gene expression omnibus (GEO) datasets GSE230932 (lung adenocarcinoma, 18 tumor/20 non-neoplastic samples) [27] and GSE120741 (prostate cancer, 49 samples) [28] containing matched RNA-seq and ChIP-seq data for histone marks (H3K4me3, H3K4me1, H3K9me3, H3K27ac for lung; H3K4me3 for prostate) were used. Similar to our previous models, we filtered the transcriptome data and used it as input features to predict histone methylation in lung and prostate cancers ([Fig 6A]). The results showed that metabolism-associated genes in lung cancer (678 genes) were able to predict histone marks with NRMSE values of 0.246 (H3K4me3), 0.307 (H3K9me3),

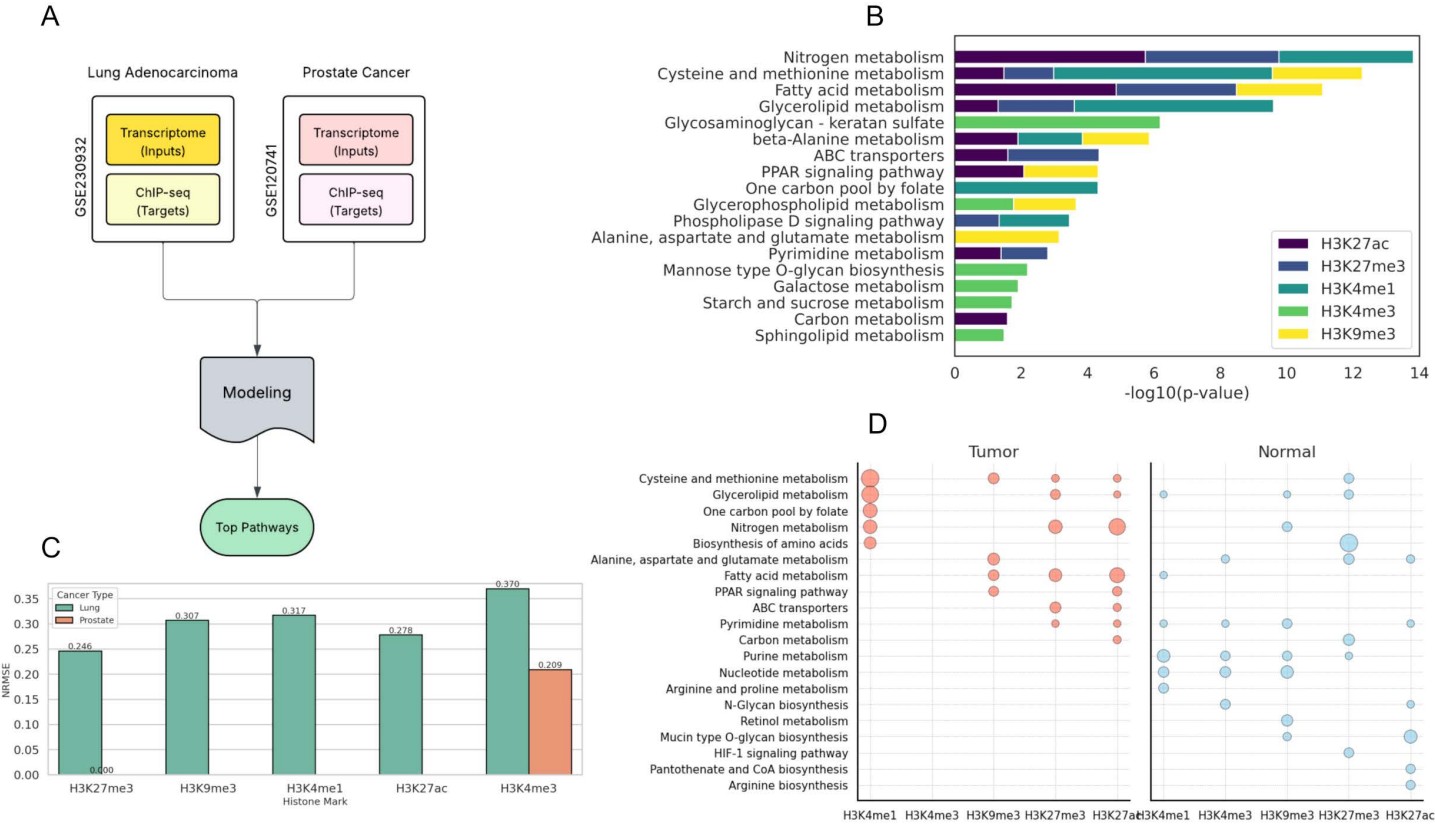

**Fig 6. Epigenetic landscape of histone modifications and associated metabolic pathways in lung adenocarcinoma and prostate cancer.** (A) Workflow diagram of the primary tumor models' pipeline for validation of CCLE models results. (B) Comparison of NRMSE values for histone modifications in primary tumors. Normalized root-mean-square error (NRMSE) values for H3K27me3, H3K9me3, H3K4me1, H3K27ac, and H3K4me3 in lung adenocarcinoma (LUAD) primary tumors and H3K4me3 in prostate primary tumors, reflecting model performance in predicting histone modification profiles. (C) DAVID pathway enrichment analysis of histone modifications in LUAD. Top enriched metabolic pathways associated with H3K27me3, H3K9me3, H3K4me1, H3K27ac, and H3K4me3 in LUAD, highlighting differential pathway regulation linked to specific histone marks. (D) Left and right panels show tumor and paired normal samples, respectively. Each panel displays histone modifications (H3K4me1, H3K4me3, H3K9me3, H3K27me3, H3K27ac) on the x-axis and enriched metabolic pathways on the y-axis. Comparative analysis highlights differences in pathway associations with specific histone marks between LUAD tumors and normal lung tissue, indicating epigenetic reprogramming of metabolism in cancer.

0.317 (H3K4me1), and 0.278 (H3K27ac), while prostate cancer features (1,628 genes) achieved an NRMSE of 0.209 (H3K4me3) (Fig 6B).

After pathway enrichment analysis on top features from these models, we observed cysteine and methionine metabolism, carbon metabolism, and TCA cycle as key predictors of H3K4me1/me2 and H3K27 marks across both cancers, consistent with the CCLE results (Fig 6C). However, the emergence of novel metabolic pathways in primary tumors, which emphasize the potential impact of the in vivo tumor microenvironment (TME) on histone methylation, highlights differences from CCLE cell line models. Notably, ABC transporters (H3K27me3, and H3K27ac) and nitrogen metabolism (H3K27ac, and H3K4me1) were among the top predictive pathways in lung adenocarcinoma, but were absent in the top features of the CCLE models. Further analysis revealed that glycosaminoglycan–keratan sulfate metabolism (H3K4me3) and fatty acid metabolism (H3K27me3, and H3K27ac) also strongly influence histone methylation in primary tumors. Results show enrichment in pathways in primary tumors such as extracellular matrix (ECM) remodeling and lipid-dependent energy or signaling demands driven by hypoxia and stromal interactions in the TME [29,30]. These TME-specific pathways were

less prominent in CCLE, likely due to homogenized nutrient availability and lack of stromal or immune components. Overall, core pathways such as cysteine/methionine and carbon metabolism were consistently identified across both cell line and primary tumor models.

## Comparison of metabolic predictors in normal and tumor tissues

To identify cancer-specific metabolic influence on histone marks, we compared the results of modeling of tumor (18 samples) and non-neoplastic (20 samples) lung tissues in GEO dataset GSE230932 (Fig 6D). Cysteine and methionine metabolism (tumor: H3K4me1, H3K9me3, H3K27me3, and H3K27ac; normal: H3K27me3), emerged as a common pathway between tumor and non-neoplastic models, likely due to its crucial role in supplying SAM for epigenetic regulation. However, tumors showed significantly stronger enrichment of this pathway across multiple histone marks. Nitrogen metabolism was also enriched in both cancer and normal tissue models, with a higher contribution in tumor cells for H3K4me1, H3K27me3, and H3K27ac, compared to the normal cells, which showed significance only in H3K9me3. This potentially suggests a cancer-specific shift toward glutamine-dependent nucleotide synthesis to support epigenetic remodeling under hypoxic TME conditions. Beyond the common pathways, we observed a shift from cellular homeostasis in normal cells to cancer-specific metabolic rewiring for proliferation and chemoresistance. Specifically, nucleotide metabolism (e.g., purine: H3K4me1) and glycan biosynthesis (e.g., mucin O-glycan: H3K27ac) were highly ranked in the normal cell models, while pathways like one-carbon pool by folate (H3K4me1), ABC transporters (H3K27me3), and PPAR signaling (H3K9me3) were highly ranked by the tumor models (Fig 6D).

## Potential mechanistic insights into regulation of histone methylation

So far, we have shown that various metabolic pathways and metabolites can partly explain histone methylation variations in cancer cells, and that the combination of omics data explains a higher level of these variations. However, it is important to note that these models do not provide directionality for associations or explain how metabolites and transcripts interact to impact methylation. One hypothesis is that perhaps variations in the cellular levels of metabolites are themselves at least in part explained by the expression levels of metabolic enzymes that generate or consume them. We next attempted to better understand this relationship computationally by modeling variations in metabolite levels using transcript levels as input features. Our goal was to depict links from cellular metabolism components (i.e., enzymes and metabolites) to histone methylation dynamics. For each metabolite in our metabolic data from the CCLE, a separate RF regression model was trained and top contributors to the cellular metabolite concentrations were determined (Fig 7A; see Methods). Consistently ranked as top features across multiple metabolites were exostosin glycosyltransferase 1 (*EXT1*), prostaglandin E synthase (*PTGES*), glutathione peroxidase 8 (*GPX8*), and MFNG O-fucosylpeptide 3-Beta-N-acetylglucosaminyltransferase (*MFNG*). Specifically, EXT1 and PTGES emerged as key predictors for 1-MNA, dimethylglycine, carnosine, and C40.6.PC, while *GPX8*, and *MFNG* were similarly associated with dimethylglycine, carnosine, and C40.6.PC, but not 1-MNA (Fig 7B). Pathway enrichment analysis revealed that top features explaining metabolite levels were concentrated in pathways such as nucleotide metabolism, glycolysis/gluconeogenesis, carbon metabolism, oxidative phosphorylation, pyruvate metabolism, and purine metabolism (Fig 7C).

## NNMT expression and clinical outcomes

The strong presence of carbon metabolism pathways (e.g., cysteine/methionine, folate, carbon metabolism) among our top contributing features for explaining histone modifications suggests a potential implication for these metabolic pathways in explaining variability in cancer pathogenesis and patient outcome. To further assess this relationship, we focused on NNMT as a key part of 1-MNA-NNMT-H3K9me2K14ac1 axis and one-carbon metabolism pathways. For this purpose, we compared survival rates between two groups (low expression versus high expression of NNMT) in each cancer

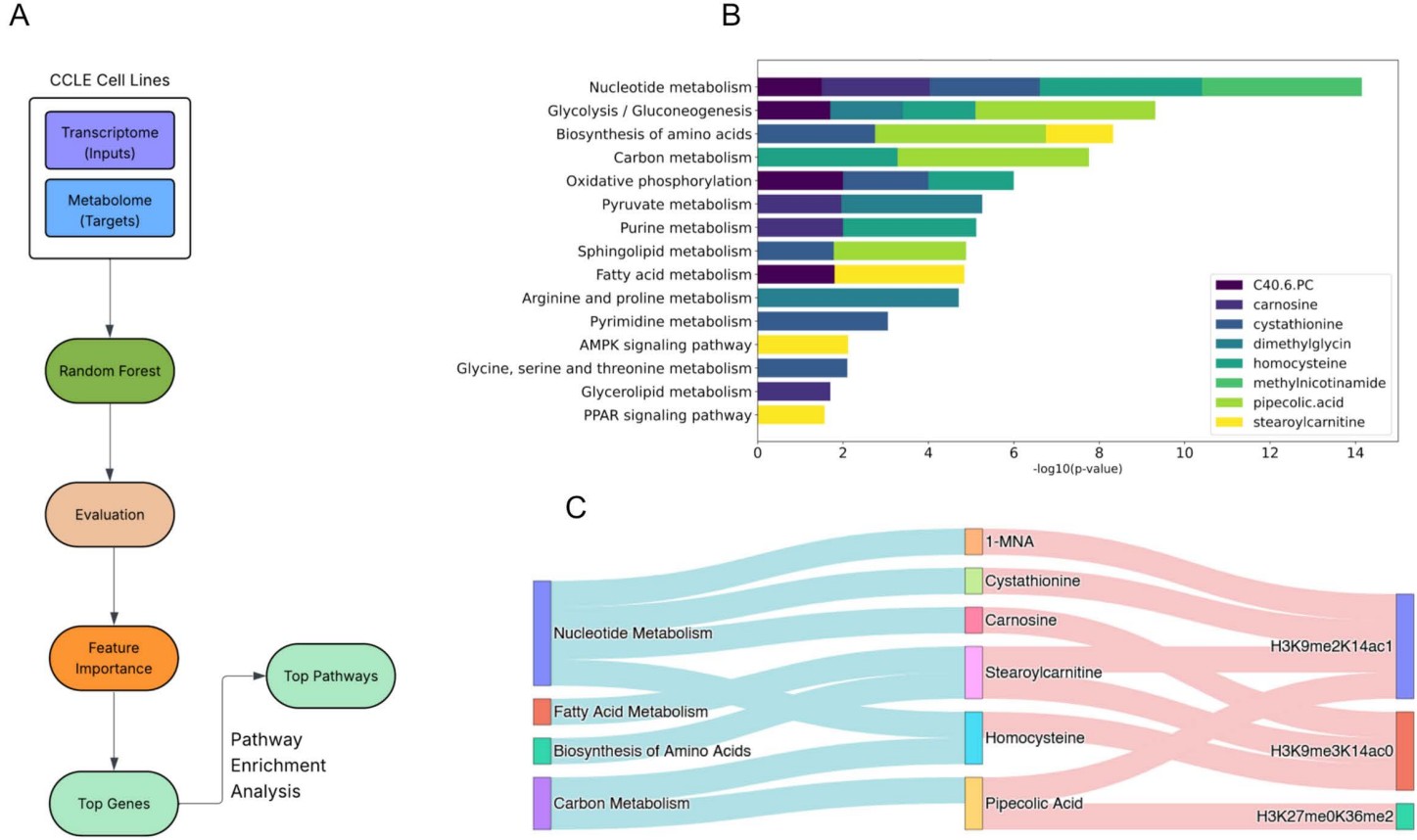

**Fig 7. Linking gene expression to metabolite variation and histone methylation.** (A) Workflow diagram of the model linking transcript levels to metabolite variations. (B) Identification of top genes influencing metabolites associated with histone methylation. (C) Integration of transcriptome and metabolome models to show potential mechanistic links between metabolic pathways, metabolites, and histone methylation.

type using the Kaplan–Meier estimator (S4A Fig). A worse overall survival was observed in the 'NNMT high expression' group, although the magnitude of this effect varied by cancer type. At a false discovery rate (FDR) ≤ 10%, significant associations were observed in stomach adenocarcinoma, head and neck squamous cell carcinoma (Head–neck SCC), and kidney renal clear cell carcinoma (Kidney RCC) (p-values: stomach = 0.0002; Head–neck SCC = 0.0004; Kidney RCC = 0.0011). In contrast, higher expression of NNMT only in sarcoma was associated with an improved overall survival (p-values = 0.0354).

We next used the TCGA data to also compare NNMT expression in cancer vs. adjacent normal tissues (S4B Fig). Notably, in bladder, breast, kidney chromophobe, liver HCC, and cholangiocarcinoma, NNMT demonstrated lower expression in comparison to normal samples (p-values: bladder = 2.55e-02, breast = 3.06e-06, kidney choromophob = 7.77e-04, liver HCC = 1.04e-07, and cholangiocarcinoma = 9.15e-03), while NNMT was overexpressed significantly in colon and kidney tumors compared to adjacent normal (RCC and Pap) (p-value: colon = 1.75e-03, Kidney RCC = 6.41e-13, Kidney Pap = 6.67e-05). This pattern aligns with our metabolic findings, where NNMT's role in SAM consumption and 1-MNA production may drive epigenetic shifts favoring tumorigenesis in overexpressing cancers (e.g., kidney RCC, where high NNMT correlates with poor survival and metabolic reprogramming for proliferation) [31]. In contrast, lower expression in cancers like liver HCC and lung SCC may accumulate nicotinamide, inhibiting Sirtuins and altering histone deacetylation [32]. The paradigm shift could stem from tissue-specific metabolic adaptations: overexpression in proliferative

environments (e.g., kidney, colon) depletes SAM to remodel epigenetics for oncogene activation, while down-regulation in others (e.g., liver, lung) may reflect compensatory responses to oxidative stress or TME nutrient limitations.

## Discussion

In this study, we used the CCLE metabolome and transcriptome datasets to assess 26 different histone methylation marks quantitatively. Machine-learning based modeling was leveraged to overcome the issues of large feature size in 'omics' data. This allowed for meaningful feature extraction and interpretation.

Our models confirmed some previously shown links as well as found potential new insights. The transcriptome models showed that genes including CAT, PIK3CA, and NUP50 are among top predictors of H3K9me2K14ac1 histone mark. Furthermore, this histone mark is linked to pathways involved in inositol and phosphatidylinositol signaling, which are integral to cellular communication and metabolism. The PIK3CA gene by encoding PI3K p110α, a key enzyme that phosphorylates phosphatidylinositol lipids, leads to PI3K/Akt activation [33]. This metabolic shift directly regulates H3K27 methylation through inhibiting EZH2 histone methyl transferase, and enhancing histone acetylation via activation of p300/CBP [34]. Furthermore, the activation of PI3K/Akt signaling pathway by influencing glycolysis, glutamine metabolism, and mitochondrial respiration, could indirectly modulate cell epigenetic landscape [35,36]. These metabolic shifts by altering the availability of key metabolite such as SAM, acetyl-CoA, and $NAD^+$ which serve as essential cofactors for chromatin-modifying enzymes [37,38]. For instance, PI3K/Akt-medicated glycolysis hyperactivation significantly alter $NAD^+$ rate and its regeneration via the salvage pathway [35,39]. Using CCA on metabolome data, we observed a significant negative correlation between 1-MNA and H3K9me2K14ac1. However, this correlation does not establish causation and could reflect confounding factors. We hypothesize that NNMT, which produces 1-MNA from NAM using SAM, may contribute to this association [40].

NNMT is the central enzyme in producing 1-MNA acting as a bridge between metabolism and epigenetic modifications. As it has been shown in multiple studies, NNMT by using SAM as a cofactor, methylates NAM and produces 1-MNA. NNMT is upregulated in cancers like esophageal squamous cell carcinoma and breast cancer, where it alters histone marks such as H3K4me3 and H3K9me3 [41,42]. However, its potential influence on H3K9me2K14ac1 as a dual mark combining repressive dimethylation (H3K9me2) and activating acetylation (H3K14ac) is less clear. Interestingly, acetylation at H3K14 has been shown to enhance SETDB1-mediated methylation of H3K9, facilitating the establishment of H3K9me3 [43]. The functional role of this mark remains poorly understood, necessitating further investigation into its biological significance. One possible mechanism involves NNMT depletion of SAM, a methyl donor for histone methyltransferases, potentially reducing H3K9me2 levels [44]. Yet, SAM is utilized by numerous cellular processes, and NNMT's impact on global SAM pools may be limited or context-specific. Additionally, by methylating NAM, NNMT could reduce NAM availability for $NAD^+$ synthesis, affecting sirtuin-mediated histone deacetylation [45]. However, alternative $NAD^+$ synthesis pathways might mitigate this effect, complicating the link to H3K9me2K14ac1. Based on our findings, the balance between NNMT and HMTs activity could modulate the availability of SAM for H3K9me2K14ac1. Due to this interconnected role of NNMT, it has a great diagnostic and therapeutical potentials [44,46,47]. For instance, NNMT knockdown demonstrated significant tumorigenesis and chemoresistance reduction, and pharmacological inhibition using the natural compound *yuanhuadine* has been reported to reverse drug resistance in lung cancer [47]. Given our results, targeting NNMT could alter the 1-MNA-H3K9me2K14ac1 axis and aberrant downstream gene regulation in cancer cells.

To address potential heterogeneity across cancer types, we analyzed certain cancer types separately, including lung, skin, hematopoietic/lymphoid tissue, and breast cancers. Cancer-type-specific analyses highlighted tissue heterogeneity as different features were found to contribute to histone methylation levels in lung cancer vs. hematopoietic/lymphoid cancers. While using the CCLE cell lines enabled us to study a comprehensive association between the metabolic network components and histone marks, the lack of tumor microenvironment in cell line may limit the findings' translational relevance. The identification of ABC transporters and pathways like glycosaminoglycan (GAG)–keratan sulfate metabolism in

the primary tumor models, highlight the crucial role of extracellular and stromal-derived metabolic cues on chromatin regulation [29,30]. In cancer cells, aberrant expression and sulfation of keratan sulfate through activation of PI3K/Akt signaling has been observed, which in turn promotes the enrichment of H3K4me3 marks at oncogenic promoters, thereby sustaining transcriptional programs involved in tumor progression and metastasis [48,49]. Likewise, ABC transporters, nitrogen metabolism and PPAR signaling pathways in primary tumors reflect adaptive responses to nutrient stress, hypoxia, and drug or metabolite efflux, which are rarely recapitulated in uniform in vitro conditions [29,50,51].

The enrichment of cysteine and methionine metabolism for H3K27me3 in both tumor and normal samples reflects the central role of this axis in histone methylation through SAM production. However, in tumor cells this metabolic pathway was recognized as the top predictor across multiple marks (H3K4me1, H3K9me3, H3K27me3, and HK27ac). Cancer cells are known to elevate methionine uptake and methionine cycle activity to sustain histone methylation and maintain transcriptional programs favoring tumor growth [24,52]. Previous studies have also shown that methionine over consumption in cancer cells allow histone methyltransferases to maintain methylation on histone marks, thereby regulating transcription toward proliferation and stemness [53–56]. Similarly, the stronger involvement of nitrogen metabolism in tumors aligns with reports that glutamine serves as a nitrogen and carbon donor for nucleotide biosynthesis and α-ketoglutarate production, influencing methylation levels through TET-mediated demethylation [57,58]. Enrichment of folate-mediated one-carbon metabolism and ABC transporter pathways in tumor cells in contrast to normal cells, potentially highlights the cancer cells' ability for metabolic adaptation to survival under redox and chemotherapeutic stress. In contrast, the maintenance of nucleotide and glycan biosynthesis signatures in normal samples suggests epigenetic regulation dominated by balanced biosynthetic demands and structural homeostasis rather than proliferative pressure. Clinically, our survival analysis results suggest that NNMT's higher expression in stomach adenocarcinoma, head and neck squamous cell carcinoma (Head–neck SCC), and kidney renal clear cell carcinoma (Kidney RCC) were associated with worse prognosis. It has been reported that NNMT's upregulation in KIRC, through PI3K/Akt/SP1/MMP-2, leads to tumor invasion and progression [44]. This emphasizes on the substantial potential of NNMT as a promising therapeutics in various cancer types, where lowering its expression could restore SAM balance, mitigate aberrant epigenetic modifications, and potentially improve clinical outcomes in cancer patients.

## Methods

### Data acquisition and preprocessing

H3 global chromatin profiling, Transcriptome [14], and Metabolome [15] data were downloaded from the CCLE (https://depmap.org/portal/ccle/). We used transcriptome, metabolome and histone methylation data consisting 870 cell lines, across 23 cancer types. This approach not only provided a large sample size to enhance statistical power and ensure generalizability but also enabled us to identify broadly applicable relationships between metabolism and epigenetics across diverse cancer types. The metabolome dataset originally contained 225 metabolite features, the transcriptome dataset included 196,519 gene expression features, and the H3 chromatin profiling dataset comprised 42 chromatin modification features. To ensure compatibility and facilitate robust statistical analysis, we first aligned the samples of the datasets by removing differences and reducing them to 870 samples across all datasets. The metabolome dataset originally contained 225 metabolite features, the transcriptome dataset included 196,519 gene expression features, and the H3 chromatin profiling dataset comprised 42 chromatin modification features. To ensure compatibility and facilitate robust statistical analysis, we first aligned the samples of the datasets by removing differences and reducing them to 870 samples across all datasets. To narrow down our analysis, we removed 16 non-methylated variables from the chromatin profiling dataset, focusing on 26 methylation marks. Additionally, focusing on metabolic pathways, we used a union of KEGG and Reactome metabolic genes to extract only those genes involved in metabolic pathways. This process leads to a reduction of 196519 genes from the transcriptome dataset to 1927 genes specifically involved in the metabolic process.

We used the missForest package (version 1.5) [59] from R, which is based on the Random Forests algorithm to fill in missing values in the datasets. To ensure comparability and facilitate further analysis, we used the StandardSclaer

module from sklearn. Preprocessing in Python to standard data values. This is a common method in statistical analysis, that aims to rescale the features with different scales, to have to have a mean of 0 and a standard deviation of 1, to ensure all variables contribute equally.

### Canonical correlation analysis (CCA)

CCA package (version 1.2.2) [60] was utilized to conduct Canonical Correlation Analysis (CCA) to identify the linear relation between metabolome and methylome dataset. Following the analysis, we used Wilk's lambda test p-value, to assess the statistical significance of the Canonical Variates (CVs). We extracted top variables from the significant CVs based on their loadings, which is the weight or coefficient of each variable in the significant CV. This approach enabled us to identify the specific histone methylation marks and metabolites that have the most influence on the correlation of metabolism and histone methylation.

### Random forests regression models

The Random Forests is an ensemble learning method that combines the predictions from multiple decision trees to provide reliable answers for classification by mode and for regression by averaging. The algorithm is frequently being used in omics studies, due to its robustness against overfitting, effectiveness on large datasets such as transcriptomics and proteomics, and its ability to capture complex non-linear relationships and higher-order interactions between features, which are prevalent in biological systems. Moreover, by providing built-in variable importance measures, which enable us to rank and select the most influential features in the models [61,62]. We used the RandomForestRegressor module from the scikit-learn library to predict 26 histone methylation marks by developing three models: the metabolome, the transcriptome, and a combination of the metabolome and the transcriptome datasets, referred to as the combined model. To build each forest, the tree size was set to 500 to train the models on 75% of our data through 10-fold cross-validation, and 25% remained data was used to assess the performance of the models on unseen data.

### Evaluation of models

In order to evaluate the regression accuracy for the models across all histone methylation marks, we utilized Normalized Root Mean Square Error (NRMSE), which compares the root mean square error to the range of observed values, providing a normalized measure of prediction error. This metric estimates how effectively a model can predict the target value by measuring the average difference between values predicted by the model and the actual values. Following the computation of the NRMSE for histone methylation marks in each of the three models, we were able to derive an overall NRMSE for each model.

Moreover, we aimed to ensure our models not only perform well on the training data but also generalize effectively to unseen data. For this purpose, we split our data into 75% training and 25% testing sets as unseen data, using the train_test_split function from the scikit-learn library. Subsequently, we applied 10-fold cross validation on the train set. This method randomly divides the train set into 10 segments, trains the models on 9 segments, and uses the remained segment to evaluate NRMSE. Next, we averaged the NRMSE obtained from each fold as the mean of the NRMSE of the train set, compared with the MSE calculated on the 25% testing set, which was not seen by the model during training. A lower NRMSE on test data means that the models not only avoided memorizing the train examples but also properly captured the main pattern of the data.

### Permutation testing for histone mark predictability

To identify the histone marks that are explainable by the input features, we used permutation_test_score module from scikit learn library [63]. This test calculates permutation-based p-value, which tests whether the observed model performance could have occurred by chance. The test generates a null distribution by randomly shuffling the labels in multiple

permutations (typically >100) and comparing the model's performance to this distribution. A low p-value (< 0.05) suggests that the model has learned a true dependency between features and labels, while a high p-value indicates that the model's performance may be due to random chance or a poor fit. This approach helps assess the robustness and validity of the model's predictions.

### Pathway enrichment analysis

We performed Pathway enrichment analysis by the Database for Annotation, Visualization, and Integrated Discovery (DAVID) tool, focusing on the KEGG database [64]. The top 30 genes were selected based on their feature importance rankings from the Random Forest Regression models for significant histone methylation marks. The enriched pathways that had a p-value < 0.05 and an association with the metabolic process were selected, which ensured the selection of pathways that were both statistically and biologically meaningful.

### Transcriptome-metabolome model

To determine the causal relations between the identified pathways and significant metabolites contributing to histone methylation, we trained RF models to predict metabolites by using transcriptome data as the input features. RF models were trained on 75% of the data, and the remaining 25% was used for testing the model. Feature importance for each metabolite was calculated to rank and identify the top genes. The DAVID tool was then used to conduct PEA on these top-ranked genes.

### Cancer-type-specific analysis

To address the heterogeneity in metabolism and transcriptomes across cancer types, we conducted a stratified analysis using the CCLE metabolome and transcriptome datasets, focusing on four cancer types: lung (164 samples), hematopoietic and lymphoid tissue (158 samples), skin, and breast cancers. ML models were employed to handle the high dimensionality of 'omics' data and identify significant histone methylation marks associated with metabolic pathways. For each cancer type, we used CCLE models to discover the relationship between metabolome/transcriptome data and 26 histone methylation marks.

### Validation in primary tumor samples

For validating our findings in primary tumor samples, we used non-small cell lung adenocarcinoma (GSE230932) [27] and prostate cancer data (GSE120741) [28] including RNA-seq and ChIP-seq datasets. However, due to substantial discrepancies in data format, measurement platforms, and feature dimensionality between CCLE and GEO datasets, direct model transfer and prediction were not feasible. Therefore, by training new RF models on the GEO datasets, and computing feature importance scores, we validated our CCLE models at the feature level. For the RNA-seq data we filtered and included only genes corresponding to the metabolic gene list from our CCLE RNA-seq data, which were then used as input features. For histone methylation marks, we averaged signal values across genomic regions for each mark per sample, creating a data frame where rows represent samples and the column contains the averaged signal value for each histone mark, used as model targets.

### Prognostic assessment of NNMT's expression

To explore the clinical value of our findings we considered NNMT gens as the main component of 1-MNA and H3K9me-2K14ac1 axis, which was highlighted by both CCA and RF models. In order to investigate the association between NNMT expression and patient survival, we utilized the KMplot web tool (https://kmplot.com) based on pan-cancer RNA-seq datasets [65]. Based on he "auto-select best cutoff" function in KMplot, patients were divided into low and high NNMT

expression groups.For comparison of NNMT expression between tumor and normal samples GEPIA web tool ([http://gepia.cancer-pku.cn](http://gepia.cancer-pku.cn)) [Single Gene Analysis – Boxplots] based on TCGA datasets were used. Boxplots were generated with statistical significance was assessed using a |Log2FC| cutoff of 1 and a p-value cutoff of 0.05 [66].

## Supporting information

**S1 Table. Overview of data modalities, sample size, and feature characteristics before and after filtering.** This table summarizes the core properties of each dataset used in the study, including metabolome, transcriptome, chromatin profiling, and the integrated multi-omics dataset (Combined). For each data type, the initial number of samples, total number of features prior to quality control, and the number and nature of retained features following filtering are reported. The feature "nature" column specifies whether the retained variables include metabolites, gene expression values, chromatin accessibility/epigenetic marks, or concatenated features from multi-layer integration. This table provides a comparative overview of data dimensionality reduction and the biological meaning of remaining features across modalities.
(DOCX)

**S2 Table. Significant canonical variates from metabolome–histone methylation canonical correlation analysis.** This table reports the canonical variates (CVs) that show statistically significant associations between metabolomic profiles and histone methylation features. For each significant canonical variate, the table includes the canonical correlation coefficient (Corr), F-statistic (F), numerator degrees of freedom (num df), denominator degrees of freedom (den df), and the associated p-value (Pr > F). These values indicate the strength and significance of the multivariate correlation structure between the two molecular layers.
(DOCX)

**S1 Fig. Top metabolite–histone methylation associations across significant canonical variates.** This figure displays the top-ranked metabolite–histone methylation pairs contributing to the six significant canonical variates (CVs) identified in the canonical correlation analysis. Metabolites and histone marks are ranked according to the absolute magnitude of their loading values. Positive correlations are shown in red and negative correlations in blue, highlighting opposing directional contributions within each CV. Only features exceeding the loading threshold were visualized to emphasize the most influential metabolic and epigenetic components.
(TIF)

**S2 Fig. Dot plot of shared metabolite–histone methylation associations identified by CCA and metabolome-only predictive models.** This dot plot visualizes the 13 metabolite–histone methylation associations that were consistently significant in both canonical correlation analysis (CCA) and the metabolome-only random forest regression (RFR) models. The y-axis represents the three histone marks identified as significant in the metabolome-only model, while the x-axis lists the corresponding metabolites. Dot color indicates the direction of association (red for positive correlations, blue for negative), and dot size is proportional to the absolute CCA loading value, reflecting the strength of feature contribution. This figure highlights robust metabolic predictors of histone methylation conserved across complementary analytical frameworks.
(TIF)

**S3 Fig. NNMT expression and H3K4me3-associated metabolic pathways across cancer types.** (A) Distribution of NNMT gene expression across multiple cancer types. The x-axis distinguishes tumor (T, red) from matched normal (N, green) samples, while the y-axis shows transcript abundance in transcripts per million (TPM). This panel illustrates overall patterns of NNMT dysregulation across cancers. (B) Comparison of the top metabolic pathways associated with H3K4me3 in lung (green) and prostate (red) cancers. Each dot represents a pathway, highlighting tissue-specific differences in metabolic regulation linked to H3K4me3 histone modifications.
(TIF)

**S4 Fig. NNMT expression and survival analysis.** (A) Kaplan–Meier overall survival curves stratified by NNMT expression levels. Patients were divided into high and low NNMT expression groups, illustrating the association between NNMT transcript abundance and overall survival across the studied cohort. (B) Comparison of NNMT gene expression between normal (N) and tumor (T) samples. This panel highlights the differential expression of NNMT in cancer versus normal tissues, emphasizing potential dysregulation in tumor cells.
(TIF)

## Author contributions

**Conceptualization:** Mohammad Rasouli Koohi.

**Data curation:** Mahya Mehrmohamadi.

**Formal analysis:** Mohammad Rasouli Koohi, Mahya Mehrmohamadi.

**Investigation:** Mahya Mehrmohamadi.

**Methodology:** Mohammad Rasouli Koohi, Mahya Mehrmohamadi.

**Project administration:** Mahya Mehrmohamadi.

**Supervision:** Mahya Mehrmohamadi.

**Validation:** Mohammad Rasouli Koohi, Mahya Mehrmohamadi.

**Visualization:** Mohammad Rasouli Koohi.

**Writing – original draft:** Mohammad Rasouli Koohi.

**Writing – review & editing:** Mohammad Rasouli Koohi, Mahya Mehrmohamadi.

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
