## [Decision Letter · Decision Letter 0]

24 Sep 2025

Dear Dr. Mehrmohamadi,

We look forward to receiving your revised manuscript.

Kind regards,

Austin W.T. Chiang

Academic Editor

PLOS ONE

**Journal Requirements:**

1. When submitting your revision, we need you to address these additional requirements. Please ensure that your manuscript meets PLOS ONE's style requirements, including those for file naming. The PLOS ONE style templates can be found at https://journals.plos.org/plosone/s/file?id=wjVg/PLOSOne_formatting_sample_main_body.pdf and https://journals.plos.org/plosone/s/file?id=ba62/PLOSOne_formatting_sample_title_authors_affiliations.pdf 2. Please note that PLOS One has specific guidelines on code sharing for submissions in which author-generated code underpins the findings in the manuscript. In these cases, we expect all author-generated code to be made available without restrictions upon publication of the work. Please review our guidelines at https://journals.plos.org/plosone/s/materials-and-software-sharing#loc-sharing-code and ensure that your code is shared in a way that follows best practice and facilitates reproducibility and reuse. 3. Thank you for uploading your study's underlying data set. Unfortunately, the repository you have noted in your Data Availability statement does not qualify as an acceptable data repository according to PLOS's standards. At this time, please upload the minimal data set necessary to replicate your study's findings to a stable, public repository (such as figshare or Dryad) and provide us with the relevant URLs, DOIs, or accession numbers that may be used to access these data. For a list of recommended repositories and additional information on PLOS standards for data deposition, please see https://journals.plos.org/plosone/s/recommended-repositories. 4. PLOS requires an ORCID iD for the corresponding author in Editorial Manager on papers submitted after December 6th, 2016. Please ensure that you have an ORCID iD and that it is validated in Editorial Manager. To do this, go to ‘Update my Information’ (in the upper left-hand corner of the main menu), and click on the Fetch/Validate link next to the ORCID field. This will take you to the ORCID site and allow you to create a new iD or authenticate a pre-existing iD in Editorial Manager. 5. Please include captions for your Supporting Information files at the end of your manuscript, and update any in-text citations to match accordingly. Please see our Supporting Information guidelines for more information: http://journals.plos.org/plosone/s/supporting-information. 6. If the reviewer comments include a recommendation to cite specific previously published works, please review and evaluate these publications to determine whether they are relevant and should be cited. There is no requirement to cite these works unless the editor has indicated otherwise. 

Reviewers' comments:

**Comments to the Author**

1. Is the manuscript technically sound, and do the data support the conclusions?

Reviewer #1: Partly

Reviewer #2: Yes

2. Has the statistical analysis been performed appropriately and rigorously?

Reviewer #1: Yes

Reviewer #2: Yes

3. Have the authors made all data underlying the findings in their manuscript fully available?

Reviewer #1: Yes

Reviewer #2: Yes

4. Is the manuscript presented in an intelligible fashion and written in standard English?

Reviewer #1: Yes

Reviewer #2: Yes

**Reviewer #1:** Summary and Significance

This manuscript investigates the relationship between cellular metabolism and histone methylation patterns in cancer cells using machine learning approaches applied to the Cancer Cell Line Encyclopedia (CCLE) dataset. The authors develop predictive models using metabolomic and transcriptomic data to identify metabolic factors that influence histone methylation marks across 870 cancer cell lines from 23 cancer types.

Strengths

- Large-scale systematic approach: The use of 870 cancer cell lines provides substantial statistical power

- Rigorous methodology: Proper cross-validation, permutation testing, and multiple model evaluation metrics

- Multi-omics integration: Combining metabolomic and transcriptomic data adds analytical depth

Major Concerns

1. Critical Gap: Lack of Clinical Validation

The most significant limitation of this study is the exclusive reliance on cell line data without validation in primary tumor samples. The authors should integrate The Cancer Genome Atlas (TCGA) database to address this critical gap, because cell lines underwent extensive adaptation that might not reflect primary tumor biology.

Specific TCGA Applications:

A) Primary tumor validation: Test the key metabolic-histone relationships identified in CCLE using TCGA's multi-omics data from >10,000 primary tumors

B) Clinical outcomes analysis: Examine whether the identified metabolic-epigenetic signatures correlate with patient survival, treatment response, or disease progression

C) Tumor heterogeneity assessment: Analyze how metabolism-histone relationships vary across tumor stages, grades, and molecular subtypes

D) Normal tissue comparison: Use TCGA normal tissue controls to determine if findings are cancer-specific or represent general biological relationships

2. Limited Novelty and Conceptual Advance

The connection between metabolism and epigenetics is well-established. The paper essentially confirms known relationships using a computational approach but provides limited new mechanistic insights. To enhance the novelty and impact of this work, the authors should leverage their computational framework to identify previously unexplored metabolite-epigenetic relationships. Their Random Forest models and feature importance analysis could be systematically applied to predict potential novel associations between lesser-studied metabolites and histone modifications. For instance, the authors could screen beyond the well-characterized one-carbon metabolism compounds, identify metabolites from other pathways (amino acid metabolism, lipid metabolism, vitamin derivatives) that show strong predictive power for specific histone marks.

The authors have developed a powerful screening tool but have primarily used it to validate existing knowledge rather than uncover novel relationships. By systematically mining their models for unexpected associations, they could significantly increase the scientific impact and novelty of their contribution. Such analysis would be particularly valuable for the research community as it could identify testable hypotheses about metabolites that have been overlooked in epigenetic studies, potentially leading to new therapeutic targets or biomarkers.

**Reviewer #2:**  1. Introduction: How does altered metabolism exactly impact histone methylation?

2. Results/Methods: what are differences between metabolome-only models and Random Forest (RF) models using metabolome, transcriptome, and combined datasets?

3. Results/Methods: how do the differences in question 2 affect identification of key metabolites influencing histone methylation?

4. Results/Methods: how do the differences in question 2 affect combined effects of metabolites and transcripts on histone methylation marks?

5. Results/Methods: what can authors provide biologically interpretable evidences as well as logic links, especially using knowledges of biochemistry and molecular biology, to explain: metabolites , gene and pathway associations with histone methylation, and potential mechanistic insights into regulation of histone methylation?

6. Discussions: can authors provide biologically interpretable evidences about other histone modifications except for histone methylation?

**Do you want your identity to be public for this peer review?** For information about this choice, including consent withdrawal, please see our Privacy Policy

Reviewer #1: No

Reviewer #2: No

---

## [Author Response · Author response to Decision Letter 1]

18 Nov 2025

Reviewer #1: Critical Gap – Lack of Clinical Validation

The most significant limitation of this study is the exclusive reliance on cell line data without validation in primary tumor samples. The authors should integrate The Cancer Genome Atlas (TCGA) database to address this critical gap, because cell lines underwent extensive adaptation that might not reflect primary tumor biology.

Specific TCGA Applications:A) Primary tumor validation: Test the key metabolic–histone relationships identified in CCLE using TCGA’s multi-omics data from >10,000 primary tumors.

We thank the reviewer for bringing this up and suggesting The Cancer Genome Atlas (TCGA) for a comprehensive source of primary tumor data for validation. We fully agree that validation in primary tumors is essential to assess the translational relevance of findings derived from cell lines. While the TCGA indeed provides extensive transcriptomics and genomics data across a wide range of primary tumor types, it lacks histone modifications and metabolomic data, which were the core of our study’s focus (metabolism–histone methylation interface).

To address this, we have now added validation using primary tumors to the revised manuscript by identifying a number of other suitable datasets from GEO. Datasets GSE230932 (lung adenocarcinoma, 18 tumor/20 non-neoplastic samples) and GSE120741 (prostate cancer, 49 samples) containing RNA-seq and ChIP-seq data for histone marks (H3K4me3, H3K4me1, H3K9me3, H3K27ac for lung; H3K4me3 for prostate) were analyzed as detailed in the revised paper. Our models trained on the GEO primary tumor data, and in the feature level we compared the findings with the CCLE results, as described in the revised manuscript. Results confirmed the core metabolic pathway’s role in histone methylation, yet highlighted a few novel pathways. These newly identified pathways emphasize the critical influence of the tumor microenvironment and in vivo metabolic context on chromatin regulation (Results, lines 309-336; Fig. 5).

B) Clinical outcomes analysis: Examine whether the identified metabolic-epigenetic signatures correlate with patient survival, treatment response, or disease progression

We appreciate the reviewer’s insightful suggestion to explore the prognostic relevance of our identified metabolic–epigenetic interactions. Given the lack of matched survival, histone methylation, and metabolomics data available to us, we conducted this analysis based on public gene expression data as available at kmplot.com. We assessed nicotinamide N-methyltransferase (NNMT) gene expression as the key enzyme for 1-MNA production. Based on our metabolome and combined models, 1-MNA was identified as the top predictor for H3K9me2K14ac1, which was also confirmed by our canonical correlation analysis (CCA). Moreover, due to this enzyme’s central role in interaction of glycolysis, NAD salvage pathway, redox and nucleotide metabolism, we selected it as the representative gene for assessing clinical outcomes. We conducted Kaplan–Meier survival analyses to compare overall survival between NNMT-high expressed group and lower expressed group across multiple cancer types. Our results showed that higher NNMT expression is significantly associated with poorer overall survival in stomach adenocarcinoma, head and neck squamous cell carcinoma, and kidney renal clear cell carcinoma (Results, lines 400-413; Supp Fig. 4a).

For the next step, using the TCGA data, we compared this gene expression between normal and tumor cells across multiple cancers. The results showed that NNMT was significantly over expressed in cancers such as colon and kidney (RCC and papillary), while it was down-regulated in liver, bladder, and cholangiocarcinoma. The differences between cancer types, may be attributed to the importance of tissue type and microenvironmental context in shaping metabolic and epigenetic interaction (Results, lines 413-428; Supp Fig. 4b).

C) Tumor heterogeneity assessment: Analyze how metabolism-histone relationships vary across tumor stages, grades, and molecular subtypes

We thank the reviewer for highlighting the importance of cancer metadata in. We fully agree that variations in tumor stage, grade, and molecular subtype can influence metabolic fluxes and histone modification dynamics. Since the TCGA was not directly useful in our study for its lack of ChIP-seq and LC-MS metabolic data, stage- or grade-specific analyses were not available to us.

Nonetheless, the CCLE and GEO models were used for assessment of cancer types and biological contexts that are now added to the revised version of the manuscript (Results, lines 273-300, Fig. 5 and 6). To study the effect of cancer type, we selected lung, hematopoietic/lymphoid, breast, and skin cancers from our CCLE cell line data since only those cancer types included a large enough sample size. We then conducted our analyses separately on each type to discover the interplay of metabolism and epigenetics. These models demonstrated the significant impact of tissue and cancer-subtype in histone methylation and metabolism landscape. For example, in lung cancer, histone marks such as H3K9me2K14ac1 and H3K27me0K36me2/3 were strongly associated with cysteine/methionine metabolism, carbon metabolism, and inositol phosphate signaling, while hematopoietic and lymphoid malignancies showed enrichment of fatty acid degradation, phospholipase D signaling, and PPAR signaling (Results, lines 273-300; Fig. 5).

Additionally, our validation analysis on lung and prostate primary tumors confirmed the effects of heterogeneity in association of metabolism to histone methylation. For instance, we compared top metabolic pathways that were enriched in H3K4me3 models for lung adenocarcinoma and prostate primary tumors. We observed that, in lung adenocarcinoma pathways such as glycoseaminoglycan - keratan sulfate, galactose metabolism, and glycerophospholipid metabolism were enriched, whereas carbon metabolism, TCA cycle, and biosynthesis of amino acids were the top predictive of H3K4me3 in prostate tumors.

D) Normal tissue comparison: Use TCGA normal tissue controls to determine if findings are cancer-specific or represent general biological relationships

We thank the reviewer for this suggestion to evaluate whether the identified metabolic–epigenetic relationships are specific to cancer or reflect broader physiological mechanisms. In order to address this critical question, we used GSE230932 dataset, which contains paired tumor (18 samples) and non-neoplastic (20 samples) lung tissues with matched RNA-seq and histone ChIP-seq data, enabling a direct and biologically relevant comparison between cancer-specific and general biological relationships.

Our models revealed that the effects of core metabolic pathways like cysteine/methionine metabolism which are crucial for methylation were conserved in the both tumor and normal samples. However, tumor samples showed a stronger enrichment of these pathways in a wider histone mark (H3K4me1, H3K9me3, H3K27me3, H3K27ac), compared to only H3K27me3 in normal samples. Beyond common pathways, ABC transporters, nitrogen metabolism, and PPAR signaling were enriched only in tumors but absent in normal tissues, potentially suggesting cancer metabolic reprogramming for adaption to hypoxia, nutrient stress, and drug resistance.(Results, lines 339-356; Fig. 6d).

2. Limited Novelty and Conceptual Advance

The connection between metabolism and epigenetics is well-established. The paper essentially confirms known relationships using a computational approach but provides limited new mechanistic insights. To enhance the novelty and impact of this work, the authors should leverage their computational framework to identify previously unexplored metabolite-epigenetic relationships. Their Random Forest models and feature importance analysis could be systematically applied to predict potential novel associations between lesser-studied metabolites and histone modifications. For instance, the authors could screen beyond the well-characterized one-carbon metabolism compounds, identify metabolites from other pathways (amino acid metabolism, lipid metabolism, vitamin derivatives) that show strong predictive power for specific histone marks.

The authors have developed a powerful screening tool but have primarily used it to validate existing knowledge rather than uncover novel relationships. By systematically mining their models for unexpected associations, they could significantly increase the scientific impact and novelty of their contribution. Such analysis would be particularly valuable for the research community as it could identify testable hypotheses about metabolites that have been overlooked in epigenetic studies, potentially leading to new therapeutic targets or biomarkers.

We appreciate the reviewer’s thoughtful comment and fully agree that identifying novel metabolite–epigenetic interactions is essential to enhance the conceptual contribution of our study. Previous studies mainly focused on assessing some of the more well-known associations between histones and metabolism individually and independently of the rest of the molecules inside cells. Using multi-omics data and handling it with a powerful tool like machine learning enabled us to have a holistic view of metabolism association with histone methylation when considering variations in hundreds of molecules at the transcriptome and metabolome levels. Thus, we would like to argue that despite the fact that one-carbon metabolism and histone methylation interplay’s already known, their re-discovery using comprehensive approaches and in the presence of other sources of variation is an important step toward confirmation of consistent and robust interaction.

Furthermore, our machine learning models and CCA analysis revealed several less known associations beyond the well-known one-carbon metabolism. For instance, stearoylcarnitine, myristoylcarnitine, phosphatidylcholines, and sphingomyelins were among the top predictors for H3K9me2K14ac1 and H3K9me3K14ac0, implicating lipid-derived redox balance and membrane signaling in chromatin regulation (Results, lines 112-166; Fig. 2b). Moreover, tryptophan metabolism emerged as a novel metabolic pathway that showed significant association with multiple histone marks (H3K4me1, H3K27me1K36me2/3), connecting NAD⁺ biosynthesis through the kynurenine pathway to histone modification (Results, lines 237-241; Fig. 4c). Furthermore, ABC transporter, nitrogen metabolism, and PPAR signaling pathways were exclusively discovered in primary tumors.

Our findings not only confirm previous well-known metabolism-epigenetics interconnections, but also hint to less explored metabolite–histone interactions across cancer types. We have clarified these insights in the Results and Discussion sections of the revised version, emphasizing how the inclusion of these new pathways strengthens the originality and translational relevance of our study.

Reviewer #2

1. Introduction: How does altered metabolism exactly impact histone methylation?

To further clarify this core concept, we have expanded the Introduction section with a more detailed explanation of the mechanisms linking altered metabolism to histone methylation. Specifically, we added the following text (Introduction, lines 30-41): " For instance, one-carbon metabolism and methionine cycle, by generating S-Adenosyl methionine (SAM) as a universal methyl donor for HMTs, have crucial roles in histone methylation level and dynamics of the methylation reaction [4]. Methionine overconsumption in cancer cells, by elevating SAM production for SETD1A/B as a histone methyltransferase, enhances methylation of H3K4me3, which leads to activation of oncogenes and tumor progression [5]. Conversely, methionine restriction, either by dietary interventions or gene therapy reduces SAM level, and increases S-Adenosyl homocysteine (SAH) which acts as a competitor for HMTs, thereby directly decreasing H3K4me3, H3K9me2, and H3K27me3, downregulating growth-promoting genes, and arresting cancer cells in S/G2 phase [6]. Emerging evidence suggests that, alteration of other metabolic pathways including nucleotide metabolism can impact SAM availability and change histone methylation landscape [7]. Beyond that, elevated rates of glycolysis in many cancer cells may deplete NAD+ pools, which serves as a co-factor for NAD+-dependent histone deacetylases such as Sirtuins [8,9]. " This addition provides a clearer, evidence-based overview while integrating relevant references from our study.

2. Results/Methods: what are differences between metabolome-only models and Random Forest (RF) models using metabolome, transcriptome, and combined datasets?

We thank the reviewer for this question and the opportunity to clarify the modeling framework. In our study, all models were based on Random Forest (RF) regression, but differed in the input feature sets used to predict histone methylation marks.

Metabolome-only model

Feature input: We used targeted metabolomics data from CCLE database including 225 different metabolites across 870 cell lines. The metabolome-only model aimed to predict variation in histone marks based on the metabolites available in the dataset, and also discover the most predictive metabolite that contribute to variations in histone modification levels. (Results, lines 78-84 and 111-164; Fig. 2).

Transcriptome-only model

Feature input: We used transcriptomic data from CCLE database including 1927 transcripts with metabolism-associated functions based on KEGG and Reactome annotations across 870 cell lines.

The transcriptome-only model aimed to first discover top genes, and then via pathway enrichment analysis top metabolic pathways whose expression levels were associated with histone methylation marks. This model outperformed the metabolome model, accurately predicting 19 histone marks with lower average NRMSE (~0.10). This improvement reflects that gene expression more comprehensively represents metabolic pathway activity and regulation of chromatin modifiers (Results, 173-207; Fig. 3).

Combined model

Feature input: We used the transcriptome data with 1927 transcripts and integrated it with the metabolome data including 225 metabolites, achieving 2,152 total features.

The combined model aimed capture complementary biological information from both omics layers. (Results, lines 220-258; Fig. 4).

3. Results/Methods: how do the differences in question 2 affect identification of key metabolites influencing histone methylation?

We thank the reviewer for this thoughtful question. The distinction among the three Random Forest (RF) model types directly shaped the identification and interpretation of key metabolites associated with histone methylation marks.

In the metabolome-only model we used metabolite concentrations (225 LC–MS features) to directly investigate the biochemical effects on histone methylation marks. This enabled us to identify 1-MNA, dimethylglycine, cystathionine, α-glycerophosphate, and stearoylcarnitine as the top predictors for marks such as H3K9me2K14ac1 and H3K9me3K14ac0 (Results, lines 78-84 and 111-164; Fig. 2b). These metabolites are majorly linked to canonical (one-carbon metabolism) and noncanonical pathways (lipid, redox, and nucleotide metabolism), showing that metabolite level alone can predict histone modifications in cancer cells.

When we integrated the metabolome layer with the transcriptome in the combined model, metabolite importance scores were re-weighted relative to co-expressed metabolic enzymes and pathways. This redefining due to inclusion of their transcriptional regulation, allowed to discover novel metabolic pathways that previous models were unable to predict (Results, lines 220-258; Fig. 4). Overall, differences in model design helped in better biological interpretation and strengthening confidence of our findings.

4. Results/Methods: how do the differences in question 2 affect combined effects of metabolites and transcripts on histone methylation marks?

We thank the reviewer for this important question. The integration of metabolome and transcriptome data in the combined RF models provided a more comprehensive understanding of how metabolic states and gene expression are not independent and

---

## [Decision Letter · Decision Letter 1]

30 Dec 2025

Computational assessment of the relationship between metabolism and histone methylation in cancer cells

PONE-D-25-39487R1

Dear Dr. Mehrmohamadi,

We’re pleased to inform you that your manuscript has been judged scientifically suitable for publication and will be formally accepted for publication once it meets all outstanding technical requirements.

Kind regards,

Austin W.T. Chiang

Academic Editor

PLOS One

Additional Editor Comments (optional):

Reviewers' comments:

Reviewer's Responses to Questions

**Comments to the Author**

Reviewer #1: All comments have been addressed

Reviewer #2: All comments have been addressed

2. Is the manuscript technically sound, and do the data support the conclusions?

Reviewer #1: Yes

Reviewer #2: Yes

3. Has the statistical analysis been performed appropriately and rigorously?

Reviewer #1: Yes

Reviewer #2: Yes

4. Have the authors made all data underlying the findings in their manuscript fully available?

Reviewer #1: Yes

Reviewer #2: Yes

5. Is the manuscript presented in an intelligible fashion and written in standard English?

Reviewer #1: Yes

Reviewer #2: Yes

Reviewer #1: (No Response)

Reviewer #2: Reviewer #1

C) Tumor heterogeneity assessment: Analyze how metabolism-histone relationships vary across tumor stages, grades, and molecular subtypes————this means: “they are NOT different tumor types” but difference among 1 tumor type

Question A) B) D) and “2. Limited Novelty and Conceptual Advance”:

# All above questions have been answered by Authers

Reviewer #2

1. Introduction: How does altered metabolism exactly impact histone methylation?

2. Results/Methods: what are differences between metabolome-only models and Random Forest (RF) models using metabolome, transcriptome, and combined datasets?

3. Results/Methods: how do the differences in question 2 affect identification of key metabolites influencing histone methylation?

4. Results/Methods: how do the differences in question 2 affect combined effects of metabolites and transcripts on histone methylation marks?

5. Results/Methods: what can authors provide biologically interpretable evidences as well as logic links, especially using knowledges of biochemistry and molecular biology, to explain: metabolites, gene and pathway associations with histone methylation, and potential mechanistic insights into regulation of histone methylation?

6. Discussions: can authors provide biologically interpretable evidences about other histone modifications except for histone methylation?

# All above questions have been answered by Authers

**Do you want your identity to be public for this peer review?** For information about this choice, including consent withdrawal, please see our Privacy Policy

Reviewer #1: No

Reviewer #2: No

---

## [Editor Report · Acceptance letter]

PONE-D-25-39487R1

PLOS One

Dear Dr. Mehrmohamadi,

I'm pleased to inform you that your manuscript has been deemed suitable for publication in PLOS One. Congratulations! Your manuscript is now being handed over to our production team.

Kind regards,

on behalf of

Dr. Wan-Tien Chiang

Academic Editor

PLOS One